# The use of Kudoh method for culture of *Mycobacterium tuberculosis* and *Mycobacterium africanum* in The Gambia

Tijan Jobarteh[1]*, Jacob Otu[1], Ensa Gitteh[1], Francis S. Mendy[1], Tutty Isatou Faal-Jawara[1], Boatema Ofori-Anyinam[1,4], Binta Sarr[1], Abi Janet Riley[1], Abigail Ayorinde[1], Bouke C. de Jong[2], Beate Kampmann[1], Ousman Secka[1]*, Florian Gehre[1,2,3]*

1 Medical Research Council Unit The Gambia at London School of Hygiene and Tropical Medicine, Fajara, The Gambia, 2 Institute for Tropical Medicine, Antwerp, Belgium, 3 Bernhard-Nocht-Institute for Tropical Medicine, Hamburg, Germany, 4 Rutgers New Jersey Medical School, Newark, New Jersey, United States of America

* Tijan.jobarteh@lshtm.ac.uk (TJ); gehre@bnitm.de (FG); ousman.secka@lshtm.ac.uk (OS)

## Abstract

**Data Availability Statement:** All relevant data are within the manuscript and its Supporting Information files.

### Background

*Mycobacterium tuberculosis* culturing remains the gold standard for laboratory diagnosis of tuberculosis. Tuberculosis remains a great public health problem in developing countries like The Gambia, as most of the methods currently used for bacterial isolation are either time-consuming or costly.

### Objective

To evaluate the Kudoh swab method in a West African setting in Gambia, with a particular focus on the method's performance when culturing *Mycobacterium africanum* West Africa 2 (MAF2) isolates.

### Method

75 sputum samples were collected in the Greater Banjul Area and decontaminated in parallel with both the standard N-acetyl-L-Cysteine-NaOH (NALC-NaOH) and the Kudoh swab method in the TB diagnostics laboratory in the Medical Research Council Unit The Gambia between 30th December 2017 and 25th February 2018. These samples were subsequently cultured on standard Löwenstein-Jensen and Modified Ogawa media respectively and incubated at 37°C for mycobacterial growth. Spoligotyping was done to determine if the decontamination and culture methods compared could equally detect *Mycobacterium tuberculosis*, *Mycobacterium africanum* West Africa 1 and *Mycobacterium africanum* West Africa 2.

### Result

Among the 50 smear positives, 35 (70%) were culture-positive with Kudoh and 32 (64%) were culture positive with NALC-NaOH, whilst 7(28%) of the 25 smear negative samples

**Funding:** The author(s) received no specific funding for this work.

**Competing interests:** The authors have declared that no competing interests exist.

were culture positive with both methods (Table 2). There was no significant difference in recovery between both methods (McNemar's test, p-value = 0.7003), suggesting that the overall positivity rate between the two methods is comparable. There were no differences in time-to-positivity or contamination rate between the methods. However, Kudoh yielded positive cultures that were negative on LJ and vice versa. All findings were irrespective of mycobacterial lineages.

## Conclusion

The Kudoh method has comparable sensitivity to the NALC-NaOH method for detecting *Mycobacterium tuberculosis* complex isolates. It is easy to perform and could be an add on option for mycobacterial culture in the field in The Gambia, since it requires less biosafety equipment.

## Introduction

Tuberculosis (TB) is an infectious disease caused by bacteria of the *Mycobacterium tuberculosis* complex (MTBc). Besides *M. tuberculosis* (MTB), TB in West Africa is also caused by two geographically restricted, endemic mycobacterial MTBc members, *Mycobacterium africanum* West Africa 1 (MAF1) and *M. africanum* West Africa 2 (MAF2) [1, 2]. The presence of these two lineages has implications for diagnostic assays and laboratory algorithms [1–3] as most commercially available kits were validated based on global mycobacterial strain collections lacking these two West African lineages. For instance (and in contrast to MTB lineages) strains of MAF2 have a non-functional pyruvate kinase and cannot metabolise glycerol as a sole carbon source [4]. Therefore, growth media in West Africa needs to be supplemented with pyruvate.

Smear microscopic examination is quick to perform and almost universally available, however there is a lack of sensitivity with sputum smear microscopy [5], and it cannot distinguish between live and dead bacilli in treatment monitoring. While molecular detection such as with the GeneXpert approaches the sensitivity of culture, mycobacterial culture isolation remains an important diagnostic test given the improved sensitivity [6], especially for patients with presumptive extrapulmonary TB, and for treatment monitoring of MDR-TB patients. One shortcoming of these culture methods is that they are prone to contamination. Therefore, collected sputa need to undergo thorough decontamination to eliminate other fast-growing contaminating bacteria. Globally, different mycobacterial culture methods are used with Petroff's method (adding 4% NaOH to the sputum) being one of the most commonly used procedures for mycobacterial culture [7].

In The Gambia, a modified version of the Petroff method (in which a mucolytic agent N-acetyl-L-Cysteine is added to NaOH) had been used as the standard decontamination procedure for sputum samples. This method is cumbersome, and it requires at least 45 minutes of manipulation, including use of a cooled centrifuge, by technically skilled personnel before the sample can be inoculated on LJ slopes. Furthermore, the Centers for Disease Control has recommended biosafety level 3 laboratories for this type of operations [5]. This is one of the important reasons why culturing of sputum samples, especially in rural areas with poorly equipped laboratories, is only available at the National level in The Gambia and most other developing countries.

Therefore, Kudoh and Kudoh [8] described a simplified swab method for decontamination and culturing of mycobacteria that takes only 4–5 minutes per sample. This method does not need expensive equipment such as centrifuges and biosafety cabinets, nor high level technical skills, as a swab is simply dipped into the sputum, directly and aseptically transferred into an NaOH containing tube and then (without centrifugation) inoculated on modified Ogawa medium slopes. We aimed to evaluate the Kudoh swab method in a West African setting in The Gambia, with a particular focus on the method's performance when culturing MAF2 isolates.

## Materials and methods

### Ethical consideration

The study was conducted within the framework of TB Sequel study (SCC 1523v1.1.) in the Greater Banjul area of The Gambia. TB Sequel is a multi-country study, in which Gambia is a study site. Within the established observatory TB cohort, TB Sequel investigates the clinical, microbiological, immunological, and socio-economic risk factors predicting the outcome of pulmonary TB. Participants were recruited at both healthcare facilities and the Medical Research Council (MRC) unit The Gambia and sputum samples were collected from participants [9]. Ethical approval was granted by The Gambia Government/MRC joint Ethics Committee. Study participants who could not read or write in English provided oral consent whilst those that can read and write in English provided written inform consent in accordance with the Declaration of Helsinki and were anonymized.

### Selection of sputum samples

A total of 75 sputa were examined between 30th December 2017 and 25th February 2018 in the TB diagnostics laboratory in the Medical Research Council Unit The Gambia. The TB diagnostics laboratory is a ISO15189 accredited by KENAS. Only sputum samples with a volume more than 2 ml were included in the study. Of the 75 sputa, 67% (50/75) were smear-positive. The grading of positivity was 22 (29%), 11 (15%) and 17(23%) for 1+, 2+ and 3+ respectively.1+ is when sputum contain 10–99 AFB in 100 fields, 2+ for 1–10 AFB per field (check 50 fields), and 3+ for more than 10 AFB per field (check 20 fields), respectively, using WHO standards [10]. The samples were collected from the Greater Banjul area of The Gambia. Samples were processed by a qualified laboratory Technician with a BSc in Biomedical science and more than five years' experience in a biosafety cabinet for Auramine O staining and both NALC-NaOH and Kudoh methods. The technician was blinded from patients' information like names, place of residence, etc.

### Auramine smear microscopy

Direct smear microscopy slides were prepared by taking a purulent portion of the sputum. The slides were heat fixed on a heat block, stained with 0.1% Auramine O and examined with a fluorescent microscope as previously described [11].

### NALC-NaOH method and inoculation on Löwenstein-Jensen slopes

Briefly, an equal volume of NALC-NaOH solution (4%) was added to the specimen, vortexed until liquefied (usually 5 to 20 seconds) and incubated for 15 minutes at room temperature. Phosphate buffer (pH 6.8) was added to the 50-ml level and the sample was centrifuged using a refrigerated centrifuge at 3000 X g for 20 minutes after which the supernatant was discarded. The pellet was re-suspended in 2ml sterile phosphate buffer and 250 µl were inoculated onto

each of the two LJ slopes (LJ with glycerol and LJ with pyruvate). Cultures were kept for up to 8 weeks until growth was observed at 37˚C.

## Kudoh method and inoculation on modified Ogawa medium

The Kudoh swab method was performed as reported by Kudoh and Kudoh (8). Briefly, sputum was gathered onto a sterile cotton swab, and the swab was immersed in a sterile 4% Sodium Hydroxide solution in a falcon tube for 2 minutes. The swab was removed from the falcon tube after 2 minutes and directly inoculated onto two Modified Ogawa slopes (Ogawa with glycerol and Ogawa with pyruvate) by smearing and squeezing the swab over the surface of the media [8]. Cultures were kept for up to 8 weeks until growth was observed at 37˚C.

## Evaluation of LJ and Ogawa slopes

Cultures (LJ and Ogawa) were read once a week for any growth. Colonies morphologically similar to *M. tuberculosis* as reported in [12] were identified and slides for microscopy were prepared from colonies growing on the culture media for confirmation of AFB growth. A culture was recorded as contaminated if the slant was covered by contaminating organisms at the end of 8 weeks, and no acid-fast organisms had been identified on the slant.

## Spoligotyping of isolates

A loopful of grown colonies was collected, re-suspended in 100μl of Tris-EDTA. The bacteria were centrifuged at 10,000 rpm for 15 minutes, after which the supernatant was discarded. The pellet was re-suspended in 100μl of Tris-EDTA by vortexing briefly and the tubes were placed in a heat block at 99˚C for 20 minutes, which lyses the mycobacterial cells to release the DNA in solution. In addition, the tubes were sonicated in a water bath at 100˚C for 15 minutes and spun down for 5 minutes at 14,000 rpm. The supernatant (which contains the DNA) was stored at -70˚C until ready for use.

Spoligotyping, which involves using multiple synthetic spacer oligonucleotides that are covalently bound to a membrane to obtain different hybridization patterns of the amplified DNA using the direct repeats as a target for invitro DNA amplification, was carried out for species identification from positive AFB growth as described in [13]. For identification of lineages the binary spoligotype patterns were uploaded onto the TB Lineage database.

## Data analysis

McNemar's test was used to evaluate the differences between the results of cultures processed with NALC-NaOH and Kudoh methods.

## Results

For a detailed overview of all patients' characteristics and of patients which tested positive for either of the tests please see Table 1 and S1 Table. Among the 50 smear positives, 35 (70%) were culture-positive with Kudoh and 32 (64%) were culture positive with NALC-NaOH, whilst 7(28%) of the 25 smear negative samples were culture positive with both methods (Table 2). There was no significant difference in recovery between both methods (McNemar's test, p-value = 0.7003), suggesting that the overall positivity rate between the two methods is comparable. Table 2 also shows that there is no difference in recovery between both methods in terms of new cases and follow-up samples. Only two smear positive samples were contaminated in both methods.

**Table 1. Patients characteristics of the study population and patients testing positive with either of the methods.** Samples were collected in the Greater Banjul Area / The Gambia between 30th December 2017 and 25th February 2018.

| Characteristics | Total study population | Positive with Kudoh Method | Positive with NaLC-NaOH Method |
|---|---|---|---|
| Age,Years | | | |
| 15–24 | 18 | 13 | 14 |
| 25–44 | 31 | 17 | 16 |
| 45–64 | 18 | 10 | 8 |
| ≥65 | 3 | 2 | 1 |
| Unknown | 5 | - | - |
| Sex | | | |
| Male | 52 | 31 | 30 |
| Female | 23 | 11 | 9 |
| Case | | | |
| Follow-ups (on treatment) | 17 | 11 | 9 |
| New Cases | 57 | 30 | 29 |
| Unknown | 1 | 1 | 1 |
| Smear Grades | | | |
| Positive (1+) | 22 | 15 | 14 |
| Positive (2+) | 11 | 8 | 6 |
| Positive (3+) | 17 | 12 | 12 |
| Negative (NS) | 25 | 7 | 7 |
| Total | 75 | 42 | 39 |

There was also no significant difference in time to detection between the two methods (Fig 1), as the average time to positivity for the Kudoh method was 37 days (95%CI 33.1–41.7) compared to 38 days (95%CI 33.9–42.1) for the NALC-NaOH method.

However, we observed qualitative differences between the methods. Although twenty-seven (36%) samples were culture positive with both methods, fifteen (20%) samples were culture positive with Kudoh method only (and negative with NALC-NaOH method) whilst twelve (16%) were exclusively culture positive with NALC-NaOH method only (Table 3).

To understand whether the observed qualitative performance differences could be due to culture bias of one of the two methods towards certain mycobacterial lineages (such as MAF2) we spoligotyped all mycobacterial isolates. All isolates were identified as MTB, MAF1 and MAF2; only one MAF1 was identified by both methods compared to eight MAF2 identified in both methods, four MAF2 was identify by Kudoh only and two MAF2 was by NALC-NaOH

**Table 2. Comparison of the two culture methods stratified by patient status.** Samples were collected in the Greater Banjul Area / The Gambia between 30th December 2017 and 25th February 2018.

| | Culture Methods | | | | | |
|---|---|---|---|---|---|---|
| | New Case | | Follow-Up | | Unknown | |
| Culture Outcome | Kudoh | NALC-NaOH | Kudoh | NALC-NaOH | Kudoh | NALC-NaOH |
| $S^+C^+$ | 25 | 23 | 9 | 8 | 1 | 1 |
| $S^+C^-$ (*) | 8 | 10 | 5 | 6 | - | - |
| $S^-C^+$ | 5 | 6 | 2 | 1 | - | - |
| $S^-C^-$ | 18 | 17 | - | 1 | - | - |
| $S^+C^{cdx(*)}$ | 1 | 1 | 1 | 1 | - | - |

Abbreviations: $S^+$ = smear positive; $S^-$ = smear negative; $C^+$ = culture positive; $C^-$ = culture negative; $C^{cdx}$ = culture contaminated.

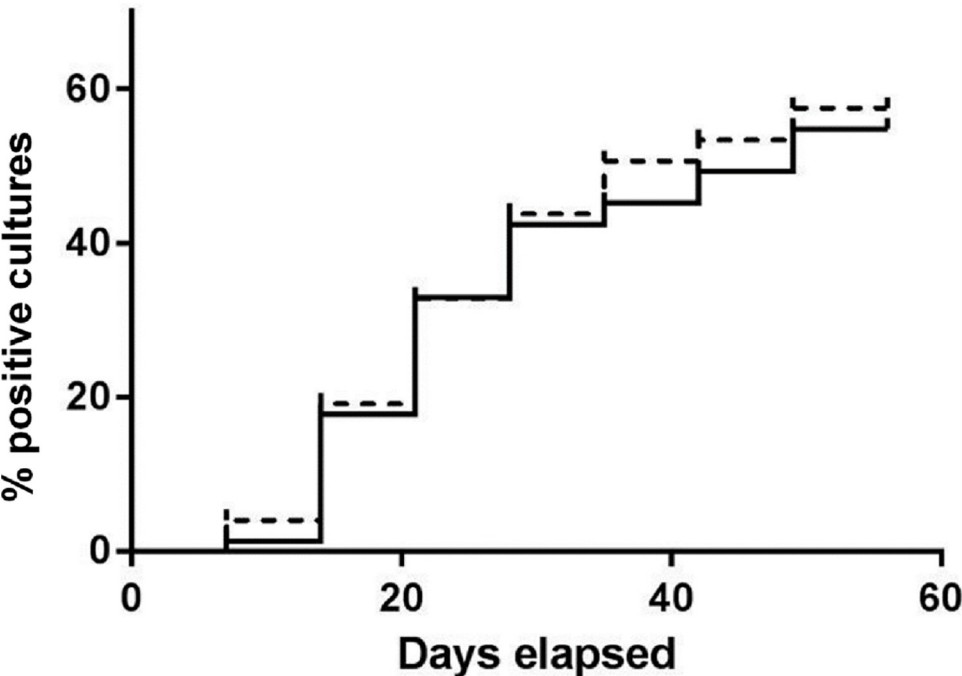

**Fig 1. Time to positivity of Kudoh modified Ogawa method (dotted line) and NALC-NaOH LJ method (solid line).**

only (Table 3). After stratifying recovery rates of each culture method by mycobacterial lineage (MAF vs. MTB), no differences were observed (p = 0.8124). Furthermore, comparing ancient (MAF, Indo-Oceanic) lineages with modern MTB lineages, no difference was observed (p = 0.6537) (Table 4).

## Discussion

We confirmed that for the diagnosis of pulmonary tuberculosis, sputum sample culture is significantly more sensitive than the direct smear.

Sputa processed with both methods in the present study were cultured on two slopes of LJ and Modified Ogawa media. Sodium pyruvate was included in both LJ and Modified Ogawa media to detect MAF based on reports that glycerol favours the growth of MTB whilst pyruvate is usually required by MAF [14].

There was no difference between both methods in mycobacterial recovery rates and contamination levels in the present study. In the present study, 42 (56%) samples were culture positive with Kudoh and 39 (52%) were culture positive with NALC-NaOH. Also, only two

**Table 3. *Mycobacterium tuberculosis* complex detection with different methods.**

| Spoligotype Family | Culture methods | | | Total |
|---|---|---|---|---|
| | Kudoh only | NALC-NaOH only | Both Kudoh & NALC-NaOH | |
| MAF 1 | - | - | 1 | 1 |
| MAF 2 | 4 | 2 | 8 | 14 |
| *M. tuberculosis* | 11 | 10 | 18 | 39 |
| Total | 15 | 12 | 27 | 54 |

**Table 4.** *Mycobacterium tuberculosis* complex lineages and family detection with different methods.

| | Kudoh | NALC-NaOH | p-value (Fisher's exact test) |
|---|---|---|---|
| Lineage | | | |
| Ancestral | 27 | 23 | 0.6537 |
| Modern | 15 | 16 | |
| Family | | | |
| MAF | 13 | 11 | 0.8124 |
| MTB | 29 | 28 | |
| Total | 42 | 39 | |

samples were contaminated in both methods in the present study. Findings from a study conducted by Rivas *et al.*, [15] in Uruguay in 2010 on Kudoh and NALC-NaOH culture methods also reported no difference regarding contamination and mycobacterial growth. Several researchers in Asian countries, including Japan, have revealed good results with the Kudoh swab method and Ogawa media [15]. Similar performances were reported when working under field conditions when the standard Petroff decontamination procedure was compared with Kudoh swab method in two Caracas hospitals, José Gregorio Hernández and José Maria Vargas in Venezuela in 2009 [16]. A recent study in Mozambique found comparable results to ours, such as exclusive detection of isolates by either method, yet at a lower magnitude, as well as a contamination rate for Ogawa method of around 4% [17].

In the present study, there was also no difference in time to detection between the two culture methods. These have also been proven in a previous study by Jaspe *et al.*, [16] in Venezuela where Kudoh and Petroff methods were compared. The authors reported no difference between the two methods in the time to detection of bacterial growth.

There was, however, a striking difference in the recovery of mycobacteria. Surprisingly, a significant number of bacteria could only be exclusively detected with either of the two methods, yet we did not see any culture bias towards a certain lineage for any of the two methods. Further studies would be needed to investigate the difference in qualitative performance in our setting and it might be advisable to include an Ogawa slope in addition to the conventional LJ slope in the routine diagnostic algorithm to increase the overall TB culture positivity rate (if solid culture is used for primary isolation of MTBc).

The Kudoh method using the Ogawa-modified medium does not require expensive equipment such as refrigerated aerosol safe centrifuges; it has a shorter turnaround time compared to standard NALC-NaOH method and relies on cheap, stable and readily available reagents [18]. Reduced equipment and fewer biosafety requirements for laboratory personnel are clear advantages of the Kudoh method and would make the method applicable for field work in The Gambia. Moreover, the Kudoh method enhances rapid culturing of sputa at field sites as it eliminates the time-consuming centrifuge steps in laboratories, thus also reducing the work load [16]. By using the Kudoh method, clinical isolates could now be recovered in areas of The Gambia where conventional NALC/LJ-based culturing was impossible to date [16]. Instead of transporting batches of sputa (in transport culture media after days of storage in the fridge) from remote areas to centralized facilities for culturing, the Kudoh-based field culturing could significantly increase recovery rates, with similar contamination rate and time to positivity, even if the slants are later sent to a central facility for incubation at 37°C [18]. This method would allow diagnosis of patients in distant areas, with no current access to TB culture capacity. Although we conducted Kudoh in combination with spoligotyping to differentiate between MAF and MTB in the present study, genotyping will not be necessary for clinical management of patients in peripheral health centres, where the Kudoh method could be implemented in the future.

## Limitations

The present study also has some limitations. Due to its small sample size, larger, prospective studies should be done in the future to confirm our results, ideally also in other West African settings where MAF is endemic. We also did not document whether the respective overall positive cultures grew on pyruvate- or glycerol-supplemented slants, yet this should be done in future studies to better understand the recovery of MAF in the Kudoh method. Finally, we did not perform spoligotype analysis on sputum samples that remained culture negative in both approaches, to identify an on overall association of culture positivity and mycobacterial lineage. Moreover, we did not test for a delay in incubation at 37°C, such as when Kudoh based inoculation would be conducted in the field and the slopes would remain at room temperature for several days before incubation.

## Conclusion

In conclusion, implementing the Kudoh method in African settings could have two benefits. First, with its acceptable sensitivity, specificity and contamination rates, Kudoh is an efficient way to culture Gambian isolates in the field, especially in regions where the NALC/LJ culture method cannot be currently done due to lack of adequate laboratory infrastructure. Secondly, we also found that some isolates were exclusively only detected with the Kudoh method, and not with NALC/LJ. Therefore, in laboratories where NALC/LJ is routinely done already, the Kudoh method could be used as an "add on" culture method to increase overall culture positivity rates by almost 20%. The routine use of the MGIT and GeneXpert system, a quick, automated method, in conjunction with the culture media (LJ and Ogawa methods) would effectively increase the diagnosis rate to control tuberculosis outbreaks. Furthermore, for the diagnosis of tuberculosis and cases involving RIF resistance, GeneXpert may be helpful. Early diagnosis of pulmonary tuberculosis is crucial for containing the spread of tuberculosis.

## Supporting information

**S1 Table.**
(XLSX)

## Acknowledgments

The authors thank all colleagues and study participants for their contributions.

## Author Contributions

**Conceptualization:** Tijan Jobarteh, Florian Gehre.

**Formal analysis:** Tijan Jobarteh, Jacob Otu, Boatema Ofori-Anyinam, Florian Gehre.

**Investigation:** Tijan Jobarteh, Florian Gehre.

**Methodology:** Tijan Jobarteh, Florian Gehre.

**Supervision:** Florian Gehre.

**Writing – original draft:** Tijan Jobarteh, Florian Gehre.

**Writing – review & editing:** Jacob Otu, Ensa Gitteh, Francis S. Mendy, Tutty Isatou Faal-Jawara, Boatema Ofori-Anyinam, Binta Sarr, Abi Janet Riley, Abigail Ayorinde, Bouke C. de Jong, Beate Kampmann, Ousman Secka, Florian Gehre.

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
