## [Decision Letter · Decision Letter 0]

6 Dec 2023

PONE-D-23-34900The use of Kudoh Method for culture of Mycobacterium tuberculosis and Mycobacterium africanum in The GambiaPLOS ONE

Dear Dr. Jobarteh,

Thank you for submitting your manuscript to PLOS ONE. After careful consideration, we feel that it has merit but does not fully meet PLOS ONE’s publication criteria as it currently stands. Therefore, we invite you to submit a revised version of the manuscript that addresses the points raised during the review process.

We look forward to receiving your revised manuscript.

Kind regards,

Padmapriya P Banada, PhD

Academic Editor

PLOS ONE

A clean copy of the edited manuscript (uploaded as the new *manuscript* file).

Additional Editor Comments:

Thank you for your submission. Please see the reviewers comments and concerns. Particularly, could you please highlight the differences in this study compared to your earlier published work (PMID: 28043531), since that study has not been cited in your current manuscript. Please also highlight and discuss in your discussion section how this new trial has contributed more to the field of study that your earlier study did not. Co-incidentally both studies seem to have same results. please explain.

I encourage the authors to answer the reviewers comments systematically. I hope these comments help the authors make the manuscript stronger.

Reviewers' comments:

Reviewer's Responses to Questions

**Comments to the Author**

1. Is the manuscript technically sound, and do the data support the conclusions?

Reviewer #1: Partly

Reviewer #2: Partly

2. Has the statistical analysis been performed appropriately and rigorously? 

Reviewer #1: Yes

Reviewer #2: Yes

3. Have the authors made all data underlying the findings in their manuscript fully available?

Reviewer #1: Yes

Reviewer #2: Yes

4. Is the manuscript presented in an intelligible fashion and written in standard English?

Reviewer #1: Yes

Reviewer #2: Yes

5. Review Comments to the Author

Reviewer #1: 1. Sample size is too small (n=75), especially in case of smear negative samples (n=25). Of the 75 sputum samples collected for evaluation only 50 are smear positive and 25 are smear negative. Need to analyse a greater number of smear positive and smear negative cases.

2.Results indicate the value of the Kudoh mycobacterial method for diagnosis of pulmonary TB as a possible alternative cost-effective method for the NALC-NaOH method. However, though rapid, of the total, seven 27 (36%) samples were culture positive with both methods, fifteen 15 (20%) samples were culture positive with Kudoh method only (and negative with NALC-NaOH method) whilst twelve 12 (16%) were exclusively culture positive with NALC-NaOH method only (Table 3). So, total culture positive specimens by both methods = 35 (positive by Kudoh method) +12 (positive by NaLC-NaOH method only) =47/75. Though the Kudoh method is comparable with NaLC-NaOH method, actually combination to both these methods Kudoh + NALC-NAOH is increasing the overall culture positivity yield by 20%. Therefore, in order to detect and treat all TB cases, in resource poor settings and /or at the peripheral sites only Kudoh method can be used as a replacement to NaLC-NaOH but at sites with better resource availability Kudoh method can be used as add on method yield of TB culture positivity by nearly 20%.

3. Once AFB culture is positive, confirmation of M. tuberculosis complex using and differentiation among M. africanum colony morphology and AFB smear are not very specific and methods like spoligotyping used for species and lineage confirmation will be technically demanding and expensive to perform at resource poor settings/regions.

Reviewer #2: The authors evaluate the Kudoh method for culture of MTB and MAF in the Gambia with a focus on recovery of MAF2 isolates.

1. The conclusion that the Kudoh method can be used as an alternative to traditional LJ methods is unfounded based on the data provided in this study. While the Kudoh method showed a slightly higher culture positivity, it still missed many MTBC cultures that were (+) by LJ (and vice versa). This translates to many patients who will be culture negative if only Kudoh methods are used. While this was briefly discussed in Lines 227-232, it is suggested that the authors modify their conclusion to focus a bit more on this limitation as well as comparison to other methods of MTBC recovery and identification (Smear, MGIT, Xpert, etc).

2. To better understand the performance between the 2 methods, it is suggested to stratify the culture results based on those with either pyruvate or glycerol (either method) to see if any increase in yield for MAF (add to Table 3). Also it is suggested to break down the cultures that were positive by each method stratified by smear grade, to determine if those with lower bacillary loads had a lower positivity on one platform vs another.

3. The paper focuses on diagnostic yield of Kudoh as well as recovery of MAF. With this in mind, the Patient characteristics as well as information concerning baseline and follow-up cultures are not needed (initially discussed in Lines 164-165, yet not in the introduction). Suggest Table 1 and 2 are re-worked to include the information RE: smear grade stratification noted in Question 2 above.

4. Note the following:

Line 15: Add "S" to State

Line 27: MAF2 should be defined or spelled out (first time in paper)

Lines 87-88: Last sentence in paragraph should be reworded, or a reference provided

Line 114: The technician was blinded to what? Please clarify.

Line 151: Please re-word, summarizing the reference noted here.

Lines 180-183: Please review and move these sentences to the appropriate sections (Materials and methods, etc)

Lines 185-187: Remove numbers "27" , "15" and "12"

Line 195: "identified"

Lines 204-205: This was not an aim of the research, please revise.

Lines 220-222: This reference found 4% of all isolates exclusively detected by either method, however the results presented here show rates at 20%(Kudoh) and 16% (NALC-NaOH) which is substantially higher. Please revise/reword.

Line 271: Reword

6. PLOS authors have the option to publish the peer review history of their article (what does this mean?). If published, this will include your full peer review and any attached files.

Reviewer #1: No

Reviewer #2: No

---

## [Author Response · Author response to Decision Letter 0]

23 Jan 2024

Rebuttal Letter for reviews to manuscript “The use of Kudoh Method for culture of Mycobacterium tuberculosis and Mycobacterium africanum in The Gambia”:

Editor Comments:

Could you please highlight the differences in this study compared to your earlier published work (PMID: 28043531), since that study has not been cited in your current manuscript. Please also highlight and discuss in your discussion section how this new trial has contributed more to the field of study that your earlier study did not. Co-incidentally both studies seem to have same results. please explain.

Response: The abstract was first submitted to the Asian-African Society of Mycobacteriology (AASM) Conference and was later approved for presentation. Regretfully, unanticipated events prevented me from physically attending the conference. Unfortunately, as the abstract was accepted, it appeared in the conference proceedings which were indexed in PubMed. This was done by the organizers without my approval, and despite my absence and the fact that I never actually presented. I recognize that this may be an uncommon situation, but I would like to reassure you that the material in our submitted manuscript is unique and hasn't been released anywhere else.

Review Comments to the Author

Reviewer #1: 1. Sample size is too small (n=75), especially in case of smear negative samples (n=25). Of the 75 sputum samples collected for evaluation only 50 are smear positive and 25 are smear negative. Need to analyse a greater number of smear positive and smear negative cases.

Response: We acknowledge the small sample size as a limitation and suggest in our “Limitation” section, that prospective, large scale studies need to be conducted in the future.

2.Results indicate the value of the Kudoh mycobacterial method for diagnosis of pulmonary TB as a possible alternative cost-effective method for the NALC-NaOH method. However, though rapid, of the total, seven 27 (36%) samples were culture positive with both methods, fifteen 15 (20%) samples were culture positive with Kudoh method only (and negative with NALC-NaOH method) whilst twelve 12 (16%) were exclusively culture positive with NALC-NaOH method only (Table 3). So, total culture positive specimens by both methods = 35 (positive by Kudoh method) +12 (positive by NaLC-NaOH method only) =47/75. Though the Kudoh method is comparable with NaLC-NaOH method, actually combination to both these methods Kudoh + NALC-NAOH is increasing the overall culture positivity yield by 20%. Therefore, in order to detect and treat all TB cases, in resource poor settings and /or at the peripheral sites only Kudoh method can be used as a replacement to NaLC-NaOH but at sites with better resource availability Kudoh method can be used as add on method yield of TB culture positivity by nearly 20%.

Response: We agree with your suggestion, and we mentioned that besides Kudoh’s advantage for field culturing, a second advantage is to use it as add-on method. We mentioned this several times in “Discussion” and “Conclusion” of the manuscript now.

3. Once AFB culture is positive, confirmation of M. tuberculosis complex using and differentiation among M. africanum colony morphology and AFB smear are not very specific and methods like spoligotyping used for species and lineage confirmation will be technically demanding and expensive to perform at resource poor settings/regions.

Response: It was not our intention to suggest that spoligotyping shall be done within routine diagnostic work to differentiate between MAF and MTB, for the very reasons the reviewer pointed out. We conducted spoligotyping in the present research study, to be able to differentiate between MAF and MTB, and to be able to compare the performance of the Kudoh method in between these two lineages in a MAF-endemic country. However, for clinical diagnosis and patient management in (peripheral) health centres, where Kudoh field culturing could now be beneficial, it is not needed to distinguish MTBC further between MAF and MTB with spoligotyping. We clarified this in the manuscript. 

Reviewer #2: The authors evaluate the Kudoh method for culture of MTB and MAF in the Gambia with a focus on recovery of MAF2 isolates.

1. The conclusion that the Kudoh method can be used as an alternative to traditional LJ methods is unfounded based on the data provided in this study. While the Kudoh method showed a slightly higher culture positivity, it still missed many MTBC cultures that were (+) by LJ (and vice versa). This translates to many patients who will be culture negative if only Kudoh methods are used. While this was briefly discussed in Lines 227-232, it is suggested that the authors modify their conclusion to focus a bit more on this limitation as well as comparison to other methods of MTBC recovery and identification (Smear, MGIT, Xpert, etc).

Response: That we suggested to replace LJ with Kudoh was a misunderstanding, and was not the intention of our statement. In order to improve our message, we modified the Discussion and Conclusion (also in line with reviewer 1’s comments) and hope that it is clearer now. The major statements that we wanted to make were (i) that Kudoh and LJ perform comparably (yet might detect different samples), (ii) that Kudoh could be a field culturing method (which cannot be done with LJ, due to all the laboratory equipment needed), and (iii) that it could be a complementary “add-on” culturing method to LJ, to increase overall culture positivity. We also expanded the discussion to mention the other TB diagnostics.

2. To better understand the performance between the 2 methods, it is suggested to stratify the culture results based on those with either pyruvate or glycerol (either method) to see if any increase in yield for MAF (add to Table 3). Also it is suggested to break down the cultures that were positive by each method stratified by smear grade, to determine if those with lower bacillary loads had a lower positivity on one platform vs another.

Response: The data to stratify culture results based on those with either pyruvate or glycerol is unfortunately not available. Although we grew each sample on the different slants, we only documented culture-positivity as the main read-out (without specifically documenting whether it was on pyruvate or glycerol). Stratification by smear grade was done, I hope is better now. We added this as a limitation in the respective section. 

3. The paper focuses on diagnostic yield of Kudoh as well as recovery of MAF. With this in mind, the Patient characteristics as well as information concerning baseline and follow-up cultures are not needed (initially discussed in Lines 164-165, yet not in the introduction). Suggest Table 1 and 2 are re-worked to include the information RE: smear grade stratification noted in Question 2 above.

Response: The change was done. 

4. Note the following:

Line 15: Add "S" to State

Response: Added S.

Line 27: MAF2 should be defined or spelled out (first time in paper)

Response: This was done.

Lines 87-88: Last sentence in paragraph should be reworded, or a reference provided

Response: The last sentence was removed.

Line 114: The technician was blinded to what? Please clarify.

Response: The technician was blinded from patients’ information like names, place of residence, etc.

Line 151: Please re-word, summarizing the reference noted here.

Response: This change was done.

Lines 180-183: Please review and move these sentences to the appropriate sections (Materials and methods, etc)

Response: This change was done.

Lines 185-187: Remove numbers "27" , "15" and "12"

Response: This change was done.

Line 195: "identified"

Response: This change was done.

Lines 204-205: This was not an aim of the research, please revise.

Response: This change was done.

Lines 220-222: This reference found 4% of all isolates exclusively detected by either method, however the results presented here show rates at 20%(Kudoh) and 16% (NALC-NaOH) which is substantially higher. Please revise/reword.

Response: This was reworded.

Line 271: Reword

Response: This change was done, I hope is better now.

---

## [Decision Letter · Decision Letter 1]

21 Feb 2024

The use of Kudoh Method for culture of Mycobacterium tuberculosis and Mycobacterium africanum in The Gambia

PONE-D-23-34900R1

Dear Dr. Jobarteh,

We’re pleased to inform you that your manuscript has been judged scientifically suitable for publication and will be formally accepted for publication once it meets all outstanding technical requirements.

Kind regards,

Padmapriya P Banada, PhD

Academic Editor

PLOS ONE

Additional Editor Comments (optional):

Thank you for answering our concerns. the manuscript indeed looks better put together. I am happy to recommend the revised manuscript for publication.

Reviewers' comments:

Reviewer's Responses to Questions

**Comments to the Author**

1. If the authors have adequately addressed your comments raised in a previous round of review and you feel that this manuscript is now acceptable for publication, you may indicate that here to bypass the “Comments to the Author” section, enter your conflict of interest statement in the “Confidential to Editor” section, and submit your "Accept" recommendation.

Reviewer #1: All comments have been addressed

2. Is the manuscript technically sound, and do the data support the conclusions?

Reviewer #1: Yes

3. Has the statistical analysis been performed appropriately and rigorously? 

Reviewer #1: Yes

4. Have the authors made all data underlying the findings in their manuscript fully available?

Reviewer #1: Yes

5. Is the manuscript presented in an intelligible fashion and written in standard English?

Reviewer #1: Yes

6. Review Comments to the Author

Reviewer #1: The authors have successfully addressed all the raised queries and this revised version is significantly improved.

7. PLOS authors have the option to publish the peer review history of their article (what does this mean?). If published, this will include your full peer review and any attached files.

Reviewer #1: **Yes: **Dr Shubhada Shenai

---

## [Editor Report · Acceptance letter]

18 Mar 2024

PONE-D-23-34900R1 

PLOS ONE

Dear Dr. Jobarteh, 

I'm pleased to inform you that your manuscript has been deemed suitable for publication in PLOS ONE. Congratulations! Your manuscript is now being handed over to our production team.

Kind regards, 

on behalf of

Dr. Padmapriya P Banada 

Academic Editor

PLOS ONE